# AGEM: Solving Linear Inverse Problems via Deep Priors and Sampling

**Bichuan Guo**
Tsinghua University
gbc16@mails.tsinghua.edu.cn

**Yuxing Han**
South China Agricultural University
yuxinghan@scau.edu.cn

**Jiangtao Wen**
Tsinghua University
jtwen@tsinghua.edu.cn

## Abstract

In this paper we propose to use a denoising autoencoder (DAE) prior to simultaneously solve a linear inverse problem and estimate its noise parameter. Existing DAE-based methods estimate the noise parameter empirically or treat it as a tunable hyper-parameter. We instead propose *autoencoder guided EM*, a probabilistically sound framework that performs Bayesian inference with intractable deep priors. We show that efficient posterior sampling from the DAE can be achieved via Metropolis-Hastings, which allows the Monte Carlo EM algorithm to be used. We demonstrate competitive results for signal denoising, image deblurring and image devignetting. Our method is an example of combining the representation power of deep learning with uncertainty quantification from Bayesian statistics.

## 1 Introduction

A variety of inverse problems, including sensor denoising [27] and image restoration [2], can be formulated as recovering a latent signal $\boldsymbol{x}$ from noisy observations $\boldsymbol{y} = H\boldsymbol{x} + \boldsymbol{n}$, where $H$ is the observation model and $\boldsymbol{n}$ is the noise. Model-based reconstruction methods [13, 20, 35] use priors to constrain the solution space. More recently, data-driven deep priors have been shown to outperform traditional analytic priors [24]. Here we adopt the unsupervised learning approach: unlike discriminative learning which requires task-specific data and training, deep priors trained with a DAE [36] can be used in a plug-and-play way [3, 4, 25], without fine-tuning for specific tasks $H$.

The noise level of $\boldsymbol{n}$ is essential for controlling the strength of prior. For example, data corrupted by large noises should be handled with strong priors. For real data, the noise level is usually unknown (i.e. noise-blind) and needs to be estimated. Although deep priors are able to capture highly sophisticated data distribution, they often lack the analytic tractability for statistical inference. As a result, many DAE-based methods either treat the noise level as a tunable hyper-parameter [3, 39], or empirically compute an adaptive estimate during gradient based optimization [4], without correctness guarantee.

In this paper, we propose a probabilistic framework that combines DAE priors with tractable inference. The latent signal $\boldsymbol{x}$ and the noise level are estimated simultaneously. We rely on the observation that a trained DAE captures the score of data distribution (gradient of log density) [1]. The key component of our method is that the intractable posterior distribution of $\boldsymbol{x}$ can be efficiently sampled with a Metropolis-Hastings [16] sampler. As a consequence, the maximum likelihood estimate (MLE) of the noise level can be obtained using the Monte Carlo EM algorithm [40]. The solution of $\boldsymbol{x}$ can be constructed from the converged samples, e.g. a minimum mean squared error (MMSE) estimator can

be computed from the posterior mean. We call our method *autoencoder guided EM* (AGEM), it is an example of marrying unsupervised deep learning with statistical inference.

One important implication of our method is that, with the aid of sampling-based approximate inference methods, a deep prior defined by a DAE can operate analytically much like closed-form priors. We demonstrate our proposed method on signal denoising, image deblurring and image devignetting, and conduct thorough ablation studies. Our approach outperforms the state-of-the-art DAE-based methods on all three tasks. In summary, the main contributions of this paper are:

- The solution of a linear inverse problem and noise level estimation are unified in a probabilistically sound framework, which can be solved using the Monte Carlo EM algorithm.

- The Monte Carlo E-step performs efficient posterior sampling with a special Metropolis-Hastings algorithm, despite using an implicit prior defined by a DAE.

- The solution to the problem can be constructed from posterior samples according to Bayesian decision theory. Using a quadratic loss, the posterior mean provides an MMSE estimator.

## 2  Background

Using the above notation, we say a linear inverse problem has a known noise level $\Sigma$, if

$$\boldsymbol{y} = H\boldsymbol{x} + \boldsymbol{n}, \ \boldsymbol{n} \sim \mathcal{N}(\boldsymbol{0}, \Sigma). \tag{1}$$

A wide range of problems can be covered by this formulation. For example, for image denoising, $H$ is the identity operator. If $\boldsymbol{x}$ is convolved with some kernel, $H$ is the Toeplitz matrix [15] of that kernel. The solution of (1) can be obtained by considering the (log) posterior distribution:

$$\log \Pr(\boldsymbol{x} \mid \boldsymbol{y}, \Sigma) = \log \Pr(\boldsymbol{y} \mid \boldsymbol{x}, \Sigma) + \log \Pr(\boldsymbol{x}) + \textit{const}. \tag{2}$$

we can view $\log \Pr(\boldsymbol{y} \mid \boldsymbol{x}, \Sigma)$ as a data term determined by model (1), and $\log \Pr(\boldsymbol{x})$ as a prior term. The data term ensures that $\boldsymbol{x}$ agrees with the observation $\boldsymbol{y}$, and the prior term regularizes $\boldsymbol{x}$ to lie in some desired solution space. For various types of data (e.g. images), many analytic priors have been proposed [17, 21, 30]. In this paper, we are interested in data-driven deep priors, as they can benefit from large amount of data and require less handcrafting. Specifically, we focus on deep priors defined by a DAE. Since a DAE uses unsupervised training, it can directly capture the probability distribution of $\boldsymbol{x}$ and does not rely on the context of task $H$ (i.e. plug-and-play), which makes it more general and widely applicable than other context-dependent priors.

**DAE prior.**  A DAE is trained to minimize the following denoising criterion:

$$\mathcal{L}_{\text{DAE}} = \mathbb{E}_{\boldsymbol{x}, \boldsymbol{\eta}}[\ell(\boldsymbol{x}, r(\boldsymbol{x} + \boldsymbol{\eta}))], \tag{3}$$

where $\ell(\cdot)$ is the loss function, $r(\cdot)$ is the reconstruction function defined by the DAE, and $\boldsymbol{\eta}$ is a stochastic noise. The expectation is taken over the discrete training set of $\boldsymbol{x}$ and the noise distribution. Besides the plug-and-play property, a DAE also provides good analytic property, as we show below.

Alain and Bengio [1] proved the theorem that if a DAE is trained with quadratic loss and isotropic Gaussian noise $\boldsymbol{\eta} \sim \mathcal{N}(0, \sigma_{\text{tr}}^2 I)$, the optimal reconstruction function $r^*(\boldsymbol{x})$ satisfies

$$r^*(\boldsymbol{x}) = \boldsymbol{x} + \sigma_{\text{tr}}^2 \nabla_{\boldsymbol{x}} \log \Pr(\boldsymbol{x}) + o(\sigma_{\text{tr}}^2), \ \text{as} \ \sigma_{\text{tr}} \to 0, \tag{4}$$

where $\Pr(\boldsymbol{x})$ is the training data distribution, and $o(\cdot)$ is the little-o notation. We see that the reconstruction error $r^*(\boldsymbol{x}) - \boldsymbol{x}$ captures the score (gradient of log density), which enables gradient-based optimization to be used for (2). With this theorem, multiple DAE-based methods for solving (1) have been proposed. DAEP [3] seeks the maximum-a-posterior (MAP) estimator

$$\boldsymbol{x}_{\text{MAP}} = \text{argmax}_{\boldsymbol{x}} \ \log \Pr(\boldsymbol{y} \mid \boldsymbol{x}, \Sigma) + \log \Pr(\boldsymbol{x}). \tag{5}$$

It uses the negative square magnitude of reconstruction error $-\|r(\boldsymbol{x}) - \boldsymbol{x}\|^2$ as a proxy prior, as it vanishes at the maxima of $\Pr(\boldsymbol{x})$. DMSP [4] proposes a Bayes estimator for a specific utility function by smoothing $\Pr(\boldsymbol{x})$ with the Gaussian kernel $\mathcal{N}(0, \sigma_{\text{tr}}^2 I)$, then makes use of an exact version (without the little-o) of (4).

**Plug-and-play ADMM.** Another DAE-based approach that does not rely on (4) originates from the fact that DAE can be used as a denoiser [8, 41]. The plug-and-play ADMM method [5] converts (5) into a constrained optimization problem:

$$(\boldsymbol{x}_{\text{MAP}}, \boldsymbol{x}_{\text{MAP}}) = \text{argmax}_{(\boldsymbol{x}, \boldsymbol{v})} \ \log \Pr(\boldsymbol{y} \mid \boldsymbol{x}, \Sigma) + \log \Pr(\boldsymbol{v}),$$
$$\text{subject to } \boldsymbol{x} = \boldsymbol{v}. \tag{6}$$

This maximizer can then be found by repeatedly solving a sequence of subproblems:

$$\text{the } \boldsymbol{x}\text{-subproblem:} \quad \boldsymbol{x}^{(k+1)} = \text{argmax}_{\boldsymbol{x}} \ \log \Pr(\boldsymbol{y} \mid \boldsymbol{x}, \Sigma) - \tfrac{\lambda}{2}\|\boldsymbol{x} - \boldsymbol{v}^{(k)} + \boldsymbol{u}^{(k)}\|^2, \tag{7}$$

$$\text{the } \boldsymbol{v}\text{-subproblem:} \quad \boldsymbol{v}^{(k+1)} = \text{argmax}_{\boldsymbol{v}} \ \log \Pr(\boldsymbol{v}) - \tfrac{\lambda}{2}\|\boldsymbol{v} - (\boldsymbol{x}^{(k+1)} + \boldsymbol{u}^{(k)})\|^2, \tag{8}$$

$$\text{update:} \quad \boldsymbol{u}^{(k+1)} = \boldsymbol{u}^{(k)} + \boldsymbol{x}^{(k+1)} - \boldsymbol{v}^{(k+1)}. \tag{9}$$

Here $\lambda$ is a positive hyper-parameter. The $\boldsymbol{x}$-subproblem (7) has an analytic solution, while the $\boldsymbol{v}$-subproblem (8) can be interpreted as a denoising step. An off-the-shelf denoiser can be used [35] to implicitly define $\Pr(\boldsymbol{v})$. Specifically, the DAE can be used to replace (8) as $\boldsymbol{v}^{(k+1)} = r(\boldsymbol{x}^{(k+1)} + \boldsymbol{u}^{(k)})$. Under mild conditions [5], the iterates (7)-(9) converge to the correct solution.

# 3  Method

The previous discussion assumes the noise level $\Sigma$ in the data term

$$\log \Pr(\boldsymbol{y} \mid \boldsymbol{x}, \Sigma) = -\tfrac{1}{2}(\boldsymbol{y} - H\boldsymbol{x})^\top \Sigma^{-1}(\boldsymbol{y} - H\boldsymbol{x}) - \tfrac{1}{2}\log|\Sigma| + \textit{const.} \tag{10}$$

to be known in advance. In particular, DAEP and ADMM require a known $\Sigma$; DMSP proposes an empirical scheme, where unknown $\Sigma$ is estimated from the current iterate of $\boldsymbol{x}$ during gradient descent. It also has to introduce a utility function that leads to Gaussian smoothed log-likelihood, causing $\Sigma$ to be overestimated (shown later in experiments). More discussions on these baselines are provided in Section A of the supplementary. Here we propose a generic algorithm for solving $\boldsymbol{x}$ and unknown $\Sigma$ simultaneously using a DAE prior. Our method is probabilistically sound.

We start by computing the MLE of $\Sigma$. Since $\boldsymbol{x}$ is a latent variable, it needs to be marginalized out:

$$\Pr(\boldsymbol{y} \mid \Sigma) = \int \Pr(\boldsymbol{y}, \boldsymbol{x} \mid \Sigma)\, \mathrm{d}\boldsymbol{x} = \int \Pr(\boldsymbol{y} \mid \boldsymbol{x}, \Sigma) \Pr(\boldsymbol{x})\, \mathrm{d}\boldsymbol{x}, \tag{11}$$

where we used the independence between $\boldsymbol{x}$ and $\Sigma$. The integral in (11) is intractable, as the prior $\Pr(\boldsymbol{x})$ is defined by a neural network (DAE). To proceed, we invoke the EM algorithm [12] to maximize the expected complete-data log-likelihood $\mathcal{Q}(\Sigma, \Sigma^{(\tau)})$:

$$\mathcal{Q}(\Sigma, \Sigma^{(\tau)}) = \mathbb{E}_{\boldsymbol{x} \sim \Pr(\boldsymbol{x}|\boldsymbol{y}, \Sigma^{(\tau)})} \log \Pr(\boldsymbol{y}, \boldsymbol{x} \mid \Sigma)$$
$$= \mathbb{E}_{\boldsymbol{x} \sim \Pr(\boldsymbol{x}|\boldsymbol{y}, \Sigma^{(\tau)})} \log \Pr(\boldsymbol{y} \mid \boldsymbol{x}, \Sigma) + \log \Pr(\boldsymbol{x}), \tag{12}$$

since the prior $\Pr(\boldsymbol{x})$ does not contain $\Sigma$, the M-step is not affected by the intractability of $\Pr(\boldsymbol{x})$. However, the E-step still needs to deal with $\Pr(x)$ as it enters the posterior distribution via

$$\Pr(\boldsymbol{x} \mid \boldsymbol{y}, \Sigma^{(\tau)}) = Z^{-1} \Pr(\boldsymbol{y} \mid \boldsymbol{x}, \Sigma^{(\tau)}) \Pr(\boldsymbol{x}), \tag{13}$$

where $Z$ is the partition function. A key component of our method is that the posterior (13) can be efficiently sampled if the prior $\Pr(\boldsymbol{x})$ is defined by a DAE, as we will show in Section 3.1. Therefore, the Monte Carlo EM algorithm can be used to compute the MLE of $\Sigma$. The E-step generates $n$ samples $\{\boldsymbol{x}^{(i)}\}_{i=1}^n$ from the posterior distribution (13), and the M-step evaluates the new $\Sigma^{(\tau+1)}$ by

$$\Sigma^{(\tau+1)} = \text{argmax}_{\Sigma} \ \sum_{i=1}^n \log \Pr(\boldsymbol{y} \mid \boldsymbol{x}^{(i)}, \Sigma) = \frac{1}{n} \sum_{i=1}^n (\boldsymbol{y} - H\boldsymbol{x}^{(i)})(\boldsymbol{y} - H\boldsymbol{x}^{(i)})^\top. \tag{14}$$

In many situations, $\Sigma$ will be constrained to be either diagonal or isotropic. In either case, the solution of (14) should be determined within the constraint. It is also straightforward to extend our analysis to the multiple-$\boldsymbol{y}$ case, where all $\boldsymbol{y}$ share the same noise level $\Sigma$. We provide discussions on these cases in Section B of the supplementary. The E-step and M-step are repeated until convergence.

## 3.1 Sampling from the posterior distribution

The posterior distribution (13) can be sampled using the Metropolis-Hastings (MH) algorithm. As we shall see, the unknown partition function $Z$ cancels out, and the theorem (4) can convert the DAE-based prior $\Pr(\boldsymbol{x})$ into tractable terms in this setting. MH requires a proposal distribution $q(\cdot \mid \boldsymbol{x}^{(i)})$. For simplicity, we first consider a Gaussian proposal $\mathcal{N}(\boldsymbol{x}^{(i)}, \sigma_{\text{prop}}^2 I)$, where $I$ is the identity matrix and $\sigma_{\text{prop}}$ is a hyper-parameter. A sample $\boldsymbol{x}^*$ is drawn from the proposal $\boldsymbol{x}^* \sim q(\cdot \mid \boldsymbol{x}^{(i)})$, and is accepted as $\boldsymbol{x}^{(i+1)} = \boldsymbol{x}^*$ with probability $\min(1, \alpha)$, where

$$\alpha = \frac{\Pr(\boldsymbol{x}^* \mid \boldsymbol{y}, \Sigma^{(\tau)}) q(\boldsymbol{x}^{(i)} \mid \boldsymbol{x}^*)}{\Pr(\boldsymbol{x}^{(i)} \mid \boldsymbol{y}, \Sigma^{(\tau)}) q(\boldsymbol{x}^* \mid \boldsymbol{x}^{(i)})}, \tag{15}$$

or otherwise rejected as $\boldsymbol{x}^{(i+1)} = \boldsymbol{x}^{(i)}$. We can rewrite (15) as

$$\log \alpha = \log \Pr(\boldsymbol{x}^* \mid \boldsymbol{y}, \Sigma^{(\tau)}) - \log \Pr(\boldsymbol{x}^{(i)} \mid \boldsymbol{y}, \Sigma^{(\tau)}) \tag{16}$$

$$= \log \Pr(\boldsymbol{y} \mid \boldsymbol{x}^*, \Sigma^{(\tau)}) - \log \Pr(\boldsymbol{y} \mid \boldsymbol{x}^{(i)}, \Sigma^{(\tau)}) + \log \Pr(\boldsymbol{x}^*) - \log \Pr(\boldsymbol{x}^{(i)}) \tag{17}$$

$$= \left( H \frac{\boldsymbol{x}^{(i)} + \boldsymbol{x}^*}{2} - \boldsymbol{y} \right)^\top \Sigma^{(\tau)-1} H(\boldsymbol{x}^{(i)} - \boldsymbol{x}^*) + \log \Pr(\boldsymbol{x}^*) - \log \Pr(\boldsymbol{x}^{(i)}), \tag{18}$$

where we used the Gaussian symmetry $q(\cdot \mid \boldsymbol{x}^*) = q(\boldsymbol{x}^* \mid \cdot)$ in the first step, the Bayes rule (13) in the second step, and the likelihood (10) in the last step. If $\boldsymbol{x}^*$ is close to $\boldsymbol{x}^{(i)}$ (e.g. $\sigma_{\text{prop}}$ is sufficiently small), we can use theorem (4) to approximate the log prior difference term in (18):

$$\log \Pr(\boldsymbol{x}^*) - \log \Pr(\boldsymbol{x}^{(i)}) \approx \nabla_{\boldsymbol{x}} \log \Pr(\boldsymbol{x}) \big|_{\boldsymbol{x}^{(i)}} \cdot (\boldsymbol{x}^* - \boldsymbol{x}^{(i)}) \tag{19}$$

$$\approx \sigma_{\text{tr}}^{-2} (r(\boldsymbol{x}^{(i)}) - \boldsymbol{x}^{(i)})^\top (\boldsymbol{x}^* - \boldsymbol{x}^{(i)}), \tag{20}$$

where the first step is a linear approximation, $r(\cdot)$ is the reconstruction function of a DAE trained with noise $\boldsymbol{\eta} \sim \mathcal{N}(0, \sigma_{\text{tr}}^2 I)$. We see that $\alpha$ can be efficiently computed using a trained DAE.

## 3.2 Efficient proposal distribution

In MH, using a fixed proposal distribution can lead to slow mixing of the Markov chain. To make sampling more efficient, the Metropolis-adjusted Langevin algorithm (MALA) [14] uses the gradient of log posterior to guide the sampler to high density regions, by adopting a special proposal $q_{\text{MALA}}$:

$$q_{\text{MALA}}(\boldsymbol{x} \mid \boldsymbol{x}^{(i)}) = \mathcal{N}(\boldsymbol{x}^{(i)} + \tfrac{1}{2}\sigma_{\text{prop}}^2 \nabla_{\boldsymbol{x}} \log \Pr(\boldsymbol{x} \mid \boldsymbol{y}, \Sigma^{(\tau)}) \big|_{\boldsymbol{x}^{(i)}}, \sigma_{\text{prop}}^2 I). \tag{21}$$

Section C of the supplementary provides some intuitions behind MALA. Interestingly, the gradient of log posterior can also be approximated using a DAE:

$$\nabla_{\boldsymbol{x}} \log \Pr(\boldsymbol{x} \mid \boldsymbol{y}, \Sigma^{(\tau)}) = \nabla_{\boldsymbol{x}} \log \Pr(\boldsymbol{y} \mid \boldsymbol{x}, \Sigma^{(\tau)}) + \nabla_{\boldsymbol{x}} \log \Pr(\boldsymbol{x}) \tag{22}$$

$$\approx H^\top \Sigma^{(\tau)-1} (\boldsymbol{y} - H\boldsymbol{x}) + \sigma_{\text{tr}}^{-2}(r(\boldsymbol{x}) - \boldsymbol{x}). \tag{23}$$

With the asymmetric proposal $q_{\text{MALA}}$, the ratio of proposals $q$ when computing $\alpha$ is no longer 1. The quantity $\log q_{\text{MALA}}(\boldsymbol{x}^{(i)} \mid \boldsymbol{x}^*) - \log q_{\text{MALA}}(\boldsymbol{x}^* \mid \boldsymbol{x}^{(i)})$, which can be readily computed from (21) and (23), needs to be added to (18) in order to evaluate the acceptance ratio $\alpha$.

## 3.3 Implementation

The previous subsections discussed how to obtain the MLE of $\Sigma$. To obtain the estimated signal $\hat{\boldsymbol{x}}$, notice that the samples drawn during the last E-step come from the posterior distribution $\Pr(\boldsymbol{x} \mid \boldsymbol{y}, \Sigma^{(\tau)})$. In principle, the Bayes estimator of common loss functions can be constructed from the posterior samples according to Bayesian decision theory (e.g. posterior mean for MSE, posterior median for L1 loss), our method is not restricted to any particular loss function. A simple choice is to use the posterior mean, which provides an MMSE estimator. The primary reason for doing so is computational: later in Table 2, we compare the posterior mean and median. Their performances are close, but the mean is easier to compute. Another reason is that many applications care about MSE (e.g. PSNR for images), hence MMSE estimator is arguably more suitable. We abbreviate this method as AGEM. Another method is to run ADMM with the estimated $\Sigma$ to obtain an MAP estimator,

---
**Algorithm 1** Estimate latent signal $\boldsymbol{x}$ and noise level $\Sigma$ with the proposed methods AGEM and AGEM-ADMM. $\tau$ is the EM iteration number, initialized as 0. $\Sigma^{(1)}$ is initialized as $\sigma_{\text{tr}}^2 I$.

---
1: Train a DAE with quadratic loss and noise $\boldsymbol{\eta} \sim \mathcal{N}(0, \sigma_{\text{tr}}^2 I)$
2: **repeat** $\tau \leftarrow \tau + 1$
3:     Initialization: If $\tau = 1$, $\boldsymbol{x}_\tau^{(1)} \leftarrow \boldsymbol{0}$, otherwise $\boldsymbol{x}_\tau^{(1)} \leftarrow \boldsymbol{x}_{\tau-1}^{(n_{\text{MH}})}$
4:     E-step: Draw $n_{\text{MH}}$ samples $\{\boldsymbol{x}_\tau^{(i)}\}_{i=1}^{n_{\text{MH}}}$ with MALA, discard the first ⅕ samples as burn-in
5:     M-step: Use $\{\boldsymbol{x}_\tau^{(i)}\}_{i=n_{\text{MH}}/5}^{n_{\text{MH}}}$ to compute $\Sigma^{(\tau+1)}$
6: **until** $\tau = n_{\text{EM}}$
7: [AGEM] Compute $\hat{\boldsymbol{x}} \leftarrow$ average of $\{\boldsymbol{x}_\tau^{(i)}\}_{i=n_{\text{MH}}/5}^{n_{\text{MH}}}$; **return** $(\hat{\boldsymbol{x}}, \Sigma^{(n_{\text{EM}})})$
8: [AGEM-ADMM] Use ADMM and noise level $\Sigma^{(n_{\text{EM}})}$ to compute $\hat{\boldsymbol{x}}$; **return** $(\hat{\boldsymbol{x}}, \Sigma^{(n_{\text{EM}})})$

---

which we abbreviate as AGEM-ADMM. Since ADMM does not depend on the approximation (4) and is based on MAP rather than MMSE, it serves as an alternative option that may perform better than AGEM. Our proposed methods are summarized in Algorithm 1. The pseudocode reflects some implementation details, which we discuss below.

**Number of iterations:** We use $n_{\text{EM}}$ to denote the total number of EM iterations, and $n_{\text{MH}}$ to denote the number of samples drawn in every E-step. We empirically find that setting $n_{\text{EM}}$ to around 20 is sufficient for convergence, meanwhile $n_{\text{MH}}$ should be large enough to achieve good mixing.

**Initialization:** MALA requires $\Sigma$ to be initialized. We empirically find that, as long as the initialization is not too far from truth, it has little impact on final results. In our implementation we initialize $\Sigma$ as the training noise $\sigma_{\text{tr}}^2 I$. As for the initial sample $\boldsymbol{x}^{(1)}$, for the first E-step we initialize it as zero; starting from the second E-step, the last sample from the previous E-step is used to initialize $\boldsymbol{x}^{(1)}$. This allows sampling to start from a high density region, rather than start from scratch.

**Burn-in:** As any MH sampler, MALA needs to run many iterations until it converges to the stationary distribution. These initial samples are discarded, known as "burn-in". In our implementation, we discard the first ⅕ samples. These discarded samples are not used in the M-step or for computing $\hat{\boldsymbol{x}}$.

The time complexity of AGEM is linear to the number of EM iterations $n_{\text{EM}}$, the number of drawn samples per iteration $n_{\text{MH}}$, and the dimension of $\boldsymbol{x}$. The space complexity of AGEM is linear to the dimension of $\boldsymbol{x}$. Note that it is not necessary to store all $n_{\text{MH}}$ samples to compute $\Sigma^{(\tau+1)}$ (line 5) or $\hat{\boldsymbol{x}}$ (line 7), as both can be computed by accumulating a partial sum, and discarding the used samples.

## 4    Related work

Noise level estimation is a crucial step for many image processing tasks, as many existing algorithms [7, 11, 29] require known noise level. Traditional noise estimation methods rely on handcrafted features or priors [17, 22, 26]. Recently, deep neural networks are used to solve a wide range of inverse problems in imaging [24]. Zhang et al. proposed CNNs for denoising [45] and super-resolution [46] that can deal with arbitrary known noise levels. In [43] they proposed denoising CNN to estimate noise levels, but their method is only applicable to an identity transformation $H = I$. Bigdeli et al. [4] proposed a deep autoencoder prior for multiple image restoration tasks with unknown noise, based on a particular utility function. Our method extend the above idea to general linear inverse problems, and we adopt the maximum likelihood principle, not limited to any subjective choices.

To simultaneously estimate the noise level $\Sigma$ and recover the latent variable $\boldsymbol{x}$, jointly maximizing the likelihood with respect to $(\boldsymbol{x}, \Sigma)$ will lead to overfitting [18]. Jin et al. [18] performed Bayes risk minimization based on a smooth utility function to prevent overfitting. A more general and objective approach is to instead marginalize out the latent variable $\boldsymbol{x}$, and perform MLE of the model parameter $\Sigma$ using the EM algorithm, as in [34, 37]. While previous work [21, 30] used tractable priors, our method performs sampling and inference with an intractable data-driven prior, combining the flexibility and representation power of deep learning with Bayesian statistics.

Our method adopts a similar philosophy as the plug-and-play ADMM literature [35]. As pointed out by [9], the ADMM method divides an MAP estimation problem into an L2 regularized inversion step and a denoising step, where the prior can be implicitly defined by an off-the-shelf denoiser [7, 11].

This allows us to use pre-trained deep architectures [6, 13, 28, 44] to overcome the limitations of traditional priors. In a similar vein, Shah and Hegde [31] proposed to use an implicit adversarial prior. A disadvantage of using implicitly defined priors is that we often lose their probabilistic interpretations, making it hard to perform model inference and requires careful parameter tuning [38]. Our framework solves this problem by using a DAE prior, which provides good analytic property.

Our method is built on the key observation by [1] that the reconstruction error of a DAE captures the score of input density. This property allows DAE to be used as image priors [32, 39, 48] to capture natural image statistics. Most relevant to us are [3, 4], where the reconstruction error is used in gradient-based optimization for image restoration. Among these, we are the first to be able to provide an MMSE estimator. Alain and Bengio [1] showed how to use MH to sample from the prior distribution defined by a DAE. Nguyen et al. [25] improved sampling in high dimensions with MALA for diverse image generation. We borrow the above ideas and show that DAE-based posterior sampling can be used in the Monte Carlo E-step to estimate model parameters.

## 5    Experimental results

We compare our approach with state-of-the-art DAE-based methods, including DMSP, DAEP, and ADMM, on various noise-blind tasks: signal denoising, image deblurring and image devignetting. We also compare to some non-DAE-based methods on specific tasks, but we do not strive for ubiquitous superior performance over task-specific methods, as the main advantage of DAE-based methods lies in their plug-and-play nature and task-agnostic generality. For each task, we train a single DAE and use it to evaluate all methods, so that they compete fairly. Since DAEP and ADMM require a noise level, we estimate it with DMSP, denoted by "DAEP+NE" and "ADMM+NE" (Noise Estimation).

All DAEs are trained by SGD with momentum 0.9 under the L2 reconstruction loss, early stopping is based on validation loss. As all baseline methods assume isotropic noise, we follow this restriction in this section for comparison purpose, and demonstrate general noise in Section E of the supplementary. For testing, $n_{\text{EM}}$ and $n_{\text{MH}}$ are set to sufficiently large values for stable convergence. We note that since the tasks are noise-blind, the hyper-parameters should not be tuned for each tested noise level. Instead, they are chosen to achieve the best $\hat{x}$ reconstruction using validation sets when $\Sigma = \sigma_{\text{tr}}^2 I$, and remain fixed for the rest experiments. Chosen values and more details are reported in each subsection. We implement and train DAEs using PyTorch [33], all experiments were run on a Ubuntu server with two Titan X GPUs. Our code and all simulated datasets will be made available online.

**Signal denoising.**    Consider 50-dimensional signals lying on a latent 2D manifold, and corrupted by isotropic Gaussian noise $\Sigma = \sigma_n^2 I$. We generate a 6000-sample dataset according to the following equation, where $\alpha, \beta \sim \text{Uniform}(2,5)$, $e = \exp(1)$, and $x_k$ is the $k$-th coordinate of the 50-dimensional signal (Section D of the supplementary provides visualization of this manifold):

$$x_k = 0.01(\alpha + \beta)^2 \sin[\alpha \sin(ke) + \beta \sin(ke + 1) + 0.5(\alpha + \beta)], k = 1, ..., 50. \qquad (24)$$

This 2D manifold is highly nonlinear. Among 6000 samples, 1000 samples are selected as the validation set and another 1000 samples as the test set. The rest are used for DAE training. The DAE is a multilayer perceptron with ReLU activations and 3 hidden layers, each containing 2000 neurons. Following [3], our DAE does not have a bottleneck as an explicit low-dimensional latent space is not required for our purpose. It is trained for 500 epochs with noise $\sigma_{\text{tr}} = 0.01$ and learning rate 0.1.

For testing, we consider four different noise levels $\sigma_n \in \{0.01, 0.02, 0.03, 0.04\}$. We compute the root-mean-square error (RMSE) between the recovered signal $\hat{x}$ and the noiseless signal $x$ by $\sqrt{\|\hat{x} - x\|^2 / 50}$, and report its mean and standard deviation (stdev.) on the test set. We set $n_{\text{EM}} = 10$, $n_{\text{MH}} = 1000$, $\sigma_{\text{prop}}$ is chosen by a grid search on $[0.001, 0.5]$. We find $\sigma_{\text{prop}} = 0.01$ achieves the best average RMSE on the validation set. Table 1 shows the results (values are scaled by 100). Our best method outperforms all baseline methods significantly statistically ($p < 0.05$), and our estimated $\sigma_n$ (in square brackets) are closer to the true values comparing to DMSP. AGEM-ADMM performs well under small noises. Indeed, since ADMM uses the trained DAE for denoising, it works well if $\sigma_n$ is close to the training noise $\sigma_{\text{tr}}$. However, as DMSP overestimates $\sigma_n$ especially when $\sigma_n$ is small, it misses the "operating region" of ADMM, leading to ADMM+NE's inferior performance.

**Ablation study.**    We study the behavior of AGEM in detail under the settings of the previous experiment. We explore different $\sigma_{\text{prop}}$, initial noise levels $\Sigma^{(1)}$, strategies to construct the recovered

Table 1: Signal denoising, average RMSE of the test set. Standard deviations are in parentheses, estimated noise levels are in square brackets. Best performances are in **bold**. (*All values are in* $10^{-2}$).

| $\sigma_n$: | 1.00 | | 2.00 | | 3.00 | | 4.00 | |
|---|---|---|---|---|---|---|---|---|
| Method | mean | std. | mean | std. | mean | std. | mean | std. |
| DAEP+NE [3] | 0.73 | (0.10) | 0.98 | (0.13) | 1.16 | (0.20) | 1.31 | (0.27) |
| ADMM+NE [35] | 0.37 | (0.28) | 0.60 | (0.36) | 0.93 | (0.55) | 1.59 | (3.49) |
| DMSP [4] | 0.50 | (0.22) | 0.74 | (0.29) | 0.99 | (0.45) | 1.36 | (0.95) |
| | [1.62] | (0.14) | [2.19] | (0.22) | [3.07] | (0.35) | [4.11] | (0.75) |
| AGEM | 0.51 | (0.15) | 0.70 | (0.25) | **0.86** | (0.39) | **1.16** | (0.64) |
| | [1.19] | (0.13) | [1.93] | (0.26) | [2.96] | (0.38) | [4.03] | (0.52) |
| AGEM-ADMM | **0.33** | (0.23) | **0.57** | (0.34) | 0.91 | (0.53) | 1.43 | (2.05) |

Table 2: Ablation study, average RMSE of the test set. Noise level is $\Sigma = \sigma_n^2 I$, where $\sigma_n = 3.00 \times 10^{-2}$. Estimated noise levels are in square brackets. (*All values are in* $10^{-2}$).

| | mean | std. | mean | std. | mean | std. | mean | std. | mean | std. |
|---|---|---|---|---|---|---|---|---|---|---|
| $\sigma_{\text{prop}}$: | 0.01 | | 0.10 | | 1.00 | | 10.0 | | 100 | |
| | 34.1 | (12.7) | 1.20 | (0.40) | 0.86 | (0.39) | does not | | does not | |
| | [34.2] | (12.7) | [2.87] | (0.40) | [2.96] | (0.38) | converge | | converge | |
| $\Sigma^{(1)}$: | 0.5$I$ | | 1.0$I$ | | 2.0$I$ | | 4.0$I$ | | 8.0$I$ | |
| | does not | | 0.86 | (0.39) | 0.87 | (0.40) | 0.86 | (0.39) | does not | |
| | converge | | [2.96] | (0.38) | [2.96] | (0.37) | [2.96] | (0.37) | converge | |
| misc. | mean | | median | | last | | first | | Gaussian | |
| | 0.86 | (0.39) | 0.87 | (0.39) | 1.57 | (0.38) | 1.58 | (0.40) | 7.61 | (4.43) |
| | [2.96] | (0.38) | [2.96] | (0.37) | [2.96] | (0.38) | [2.96] | (0.38) | [9.51] | (3.82) |

$\hat{x}$, and compare MALA with the symmetric Gaussian proposal. We set the test noise level $\sigma_n = 0.03$, all hyper-parameters remain unchanged except for the hyper-parameter being studied.

Table 2 summarizes the results (values are scaled by 100 for better display). **The first row** shows results using different $\sigma_{\text{prop}}$. If $\sigma_{\text{prop}}$ is too small, the results are incorrect, as it takes impractically many samples to achieve good mixing. If $\sigma_{\text{prop}}$ is too large, new samples deviate from high density regions, and the algorithm fails to converge as no new samples are accepted. Therefore, besides using a validation set to choose a fixed $\sigma_{\text{prop}}$, another possible strategy is to dynamically increase $\sigma_{\text{prop}}$ while keeping the algorithm convergent. We leave this for future investigation. **The second row** shows results using different noise level initializations. We see that as long as the initialization is within a good range, the results are stable. In practice one can try a wide range of initializations to seek convergence. **The third row** compares different strategies for constructing the recovered $\hat{x}$. "Mean"/"median" uses the coordinate-wise mean/median of the samples, while "last"/"first" uses the last/first sample, all from the last iteration. "Mean" and "median" achieve similar performances, while "last" and "first" have worse RMSE, as a single sample fails to represent the central tendency of the entire posterior distribution. Finally, "Gaussian" stands for using symmetric Gaussian proposal during the E-step. Comparing to "mean" which uses MALA, we see the Gaussian proposal gives incorrect results, as it fails to exploit gradient information and is stuck at local maxima.

**Image deblurring.** We perform image deblurring with the STL-10 unlabeled dataset [10], which contains $10^5$ colored 96×96 images. They are converted to grayscale and normalized to $[0, 1]$. We select the last 400 images, the first/second half of which is used as the validation/test set. The rest are used for DAE training. The DAE uses the full convolutional, residual architecture from [43], where the input is added to the final layer's output. It is trained for 250 epochs with noise $\sigma_{\text{tr}} = 0.02$ and learning rate 0.01. We empirically find that DAEs trained with smaller noises do not perform as well.

For testing, images are blurred using a $5 \times 5$ Gaussian filter with $\sigma = 0.6$. The noise is spatially uniform $\Sigma = \sigma_n^2 I$, where $\sigma_n \in \{0.01, 0.02, 0.03, 0.04\}$. We set $n_{\text{EM}} = 10$, $n_{\text{MH}} = 300$, $\sigma_{\text{prop}}$ is set to 0.02 using the same selection method as signal denoising, except RMSE is replaced by PSNR. The

Table 3: Average PSNR for image deblurring. Estimated noise levels are in square brackets.

| $\sigma_n$: | 0.01 | | 0.02 | | 0.03 | | 0.04 | |
|---|---|---|---|---|---|---|---|---|
| Method | mean | std. | mean | std. | mean | std. | mean | std. |
| DAEP+NE [3] | 33.13 | (1.39) | 27.77 | (0.89) | 25.48 | (0.70) | 24.30 | (0.61) |
| ADMM+NE [35] | 32.43 | (3.08) | 29.48 | (3.16) | 27.87 | (2.97) | 25.78 | (3.16) |
| DMSP [4] | 33.60 | (2.46) | 30.89 | (2.14) | 28.93 | (2.18) | 27.40 | (2.33) |
| | [0.017] | (1e-3) | [0.023] | (2e-3) | [0.031] | (3e-3) | [0.041] | (4e-3) |
| AGEM | **34.79** | (2.00) | **31.42** | (1.81) | **29.47** | (1.92) | 28.00 | (2.10) |
| | [0.014] | (1e-3) | [0.021] | (2e-3) | [0.030] | (3e-3) | [0.040] | (3e-3) |
| AGEM-ADMM | 33.75 | (2.77) | 30.00 | (3.20) | 28.00 | (2.88) | 26.05 | (3.51) |
| Hyper-Laplacian [21] | 33.28 | (0.65) | 30.26 | (0.40) | 29.28 | (0.35) | **28.82** | (0.35) |
| CSF [30] | 32.97 | (0.68) | 29.94 | (0.41) | 29.02 | (0.37) | 28.61 | (0.36) |

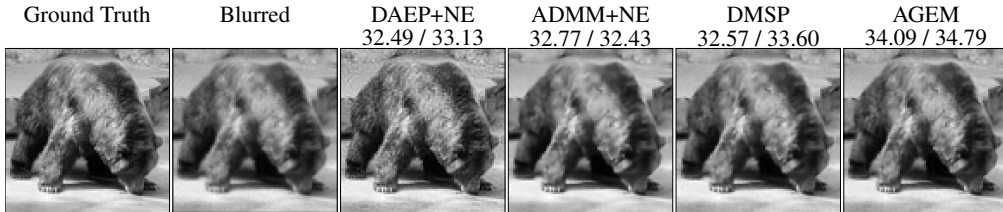

| Ground Truth | Blurred | DAEP+NE | ADMM+NE | DMSP | AGEM |
|---|---|---|---|---|---|
| | | 32.49 / 33.13 | 32.77 / 32.43 | 32.57 / 33.60 | 34.09 / 34.79 |

Figure 1: Visual comparison for image deblurring with $\sigma_n = 0.01$. Numbers above the images are: PSNR of the image / average PSNR of the test set (in dB). Zoom in for more details.

mean/stdev. of PSNR and estimated $\sigma_n$ on the test set are reported in Table 3. AGEM consistently outperforms all baseline methods significantly statistically ($p < 0.01$), and its estimated $\sigma_n$ are closer to true values than DMSP. We also compare with some analytic priors [21, 30]. Although these priors are specifically designed for image deconvolution, our generic approach outperforms them except for $\sigma_n = 0.04$, indicating that our trained DAE learns the distribution of natural images well, and DAE-based methods are indeed relevant in practice. Some visual examples are provided in Fig. 1. A convergence visualization is provided in Fig. 2, which shows the stability of our approach.

**Image devignetting.** Vignetting is a prevalent artifact in photography that brightness attenuates away from the center [47]. We perform image devignetting with the CelebA dataset [42], which contains 0.2 million $218\times178$ colored face images, and a predefined train/val/test split. We normalize images to $[0, 1]$ and train a DAE with the entire training set. We use the same DAE architecture as image deblurring. It is trained for 125 epochs with noise $\sigma_{\text{tr}} = 0.02$ and learning rate 0.1.

We select the first 100 images from the predefined val/test set as our validation/test set. The transformation is based on the Kang-Weiss [19] vignetting model

$$p(r) = \frac{1 - \alpha r}{[1 + (r/f)^2]^2}. \tag{25}$$

The intensity of a pixel, whose distance to the center is $r$, is multiplied by $p(r)$. We set $\alpha = 0.001$, $f = 160$ to achieve a realistic vignetting effect. $H$ is then a diagonal matrix if images are reshaped into

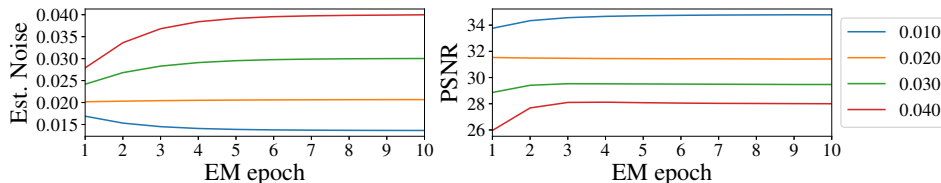

Figure 2: Convergence visualization for image deblurring. Left: average estimated noise level; right: mean PSNR. The legend shows the true noise level $\sigma_n$. Stable convergence is quickly reached. Each EM epoch draws 300 MCMC samples.

Table 4: Average PSNR for image devignetting. Estimated noise levels are in square brackets.

| $\sigma_n$: | 0.015 | | 0.02 | | 0.025 | | 0.03 | |
|---|---|---|---|---|---|---|---|---|
| Method | mean | std. | mean | std. | mean | std. | mean | std. |
| DAEP+NE [3] | 33.76 | (0.71) | 31.19 | (0.69) | 29.16 | (0.64) | 27.51 | (0.58) |
| ADMM+NE [35] | 34.10 | (1.62) | 32.95 | (1.56) | 31.60 | (1.56) | 29.96 | (1.68) |
| DMSP [4] | 35.78 | (0.99) | 34.43 | (0.94) | 33.26 | (0.94) | 32.18 | (1.03) |
| | [0.022] | (1e-3) | [0.024] | (1e-3) | [0.027] | (1e-3) | [0.032] | (1e-3) |
| AGEM | **36.34** | (0.65) | **34.76** | (0.68) | **33.58** | (0.77) | **32.55** | (0.88) |
| | [0.017] | (1e-3) | [0.020] | (1e-3) | [0.024] | (1e-3) | [0.029] | (1e-3) |
| AGEM-ADMM | 36.16 | (1.54) | 34.56 | (1.53) | 32.87 | (1.54) | 31.07 | (1.60) |
| LIE [23] | 29.61 | (1.72) | 29.43 | (1.43) | 29.23 | (1.16) | 29.05 | (0.95) |
| SIVC [47] | 29.55 | (0.87) | 29.44 | (0.78) | 29.33 | (0.71) | 29.22 | (0.64) |

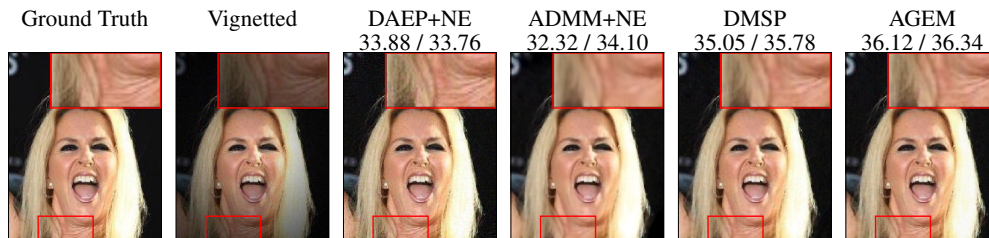

Figure 3: Visual comparison for image devignetting with $\sigma_n = 0.015$. Numbers above the images are: PSNR of the image / average PSNR of the test set (in dB). Zoom in for more details.

column vectors. We consider spatially uniform $\Sigma = \sigma_n^2 I$, where $\sigma_n \in \{0.015, 0.02, 0.025, 0.03\}$. We set $n_{\text{EM}} = 10$, $n_{\text{MH}} = 200$, $\sigma_{\text{prop}}$ is set to 0.02 using the same selection method as image deblurring. The mean/stdev. of PSNR and estimated $\sigma_n$ on the test set are reported in Table 4. AGEM consistently outperforms all baseline methods significantly statistically ($p < 0.01$), and its estimated $\sigma_n$ are closer to true values than DMSP. We also compare with existing methods [23, 47] that do not rely on the known model $p(r)$. They are outperformed by model-based methods, as $p(r)$ contains essential information for reconstruction performance. Some visual examples are provided in Fig. 3.

## 6    Concluding remarks

In this paper, we propose a probabilistic framework that uses DAE prior to simultaneously solve linear inverse problems and estimate noise levels, based on the Monte Carlo EM algorithm. We show that during the Monte Carlo E-step, efficient posterior sampling can be performed, as the reconstruction error of DAE captures the gradient of log prior. Our framework allows us to use deep priors trained by unsupervised learning for a wide range of tasks, including signal denoising, image deblurring and image devignetting. Experimental results show that our method outperforms the previous state-of-the-art DAE-based methods. However, this study is not without limitations. Since our method is based on sampling, it usually takes several times longer than non-sampling-based methods to achieve stable convergence. A possible direction for future research is to extend our framework to nonlinear inverse problems. We are also considering using other forms of deep priors.

### Acknowledgments

This work is supported by the Natural Science Foundation of China (Project Number 61521002). We would like to thank Xinyue Liang for discussions on MCMC methods. We also thank the reviewers and the area chair for their valuable comments.

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
