[Supplementary Material]

# Supplementary Material – AGEM: Solving Linear Inverse Problems via Deep Priors and Sampling

**Bichuan Guo**
Tsinghua University
gbc16@mails.tsinghua.edu.cn

**Yuxing Han**
South China Agricultural University
yuxinghan@scau.edu.cn

**Jiangtao Wen**
Tsinghua University
jtwen@tsinghua.edu.cn

## A    Generalizing baseline methods

Assume the same transformation model $\boldsymbol{y} = H\boldsymbol{x} + \boldsymbol{n}$. The baseline methods we compared in the main paper, namely DAEP [*S1*], DMSP [*S2*], and ADMM [*S4*], all assumed isotropic (spatially uniform) noise in their original papers. Here we show how to generalize them to any noise covariance matrix $\boldsymbol{n} \sim \mathcal{N}(0, \Sigma)$. We later compare these generalized baselines with AGEM in Section E. We also analyze the limitation of DMSP mathematically.

### A.1    DMSP

For DMSP, the original paper only discussed noise estimation in the context of noise-blind image deblurring. Here we generalize it to general linear inverse problems. DMSP uses a Gaussian smoothed data term (log-likelihood), as follows (page 4 of [*S2*]):

$$\text{data}(\boldsymbol{x}) = \int g_\sigma(\boldsymbol{\epsilon}) \log \Pr(\boldsymbol{y} \mid \boldsymbol{x} + \boldsymbol{\epsilon}) \, \mathrm{d}\boldsymbol{\epsilon}, \tag{S1}$$

where $g_\sigma(\cdot)$ is the probability density function (pdf) of a Gaussian distribution $\mathcal{N}(\mathbf{0}, \sigma^2 I)$. We can simplify (S1) as

$$\text{data}(\boldsymbol{x}) = \mathbb{E}_{\boldsymbol{\epsilon} \sim \mathcal{N}(\mathbf{0}, \sigma^2 I)} \ \log \Pr(\boldsymbol{y} \mid \boldsymbol{x} + \boldsymbol{\epsilon}) \tag{S2}$$

$$= \mathbb{E}_{\tilde{\boldsymbol{x}} \sim \mathcal{N}(\boldsymbol{x}, \sigma^2 I)} \ \log \Pr(\boldsymbol{y} \mid \tilde{\boldsymbol{x}}) \tag{S3}$$

$$= \mathbb{E}_{\tilde{\boldsymbol{x}} \sim \mathcal{N}(\boldsymbol{x}, \sigma^2 I)} \ -\frac{1}{2}(\boldsymbol{y} - H\tilde{\boldsymbol{x}})^\top \Sigma^{-1}(\boldsymbol{y} - H\tilde{\boldsymbol{x}}) - \frac{1}{2}\log|\Sigma| + \textit{const.} \tag{S4}$$

$$= -\frac{1}{2}\text{trace}(\mathbb{E}_{\tilde{\boldsymbol{x}} \sim \mathcal{N}(\boldsymbol{x}, \sigma^2 I)} \ \Sigma^{-1}(\boldsymbol{y} - H\tilde{\boldsymbol{x}})(\boldsymbol{y} - H\tilde{\boldsymbol{x}})^\top) - \frac{1}{2}\log|\Sigma| + \textit{const.} \tag{S5}$$

$$= -\frac{1}{2}(\boldsymbol{y} - H\boldsymbol{x})^\top \Sigma^{-1}(\boldsymbol{y} - H\boldsymbol{x}) - \frac{1}{2}\sigma^2 \text{trace}(\Sigma^{-1} H H^\top) - \frac{1}{2}\log|\Sigma| + \textit{const.}, \tag{S6}$$

where $\text{trace}(\cdot)$ is the trace operator of a matrix. We see that the gradient of the data term is simply $\nabla_{\boldsymbol{x}}\text{data}(\boldsymbol{x}) = H^\top \Sigma^{-1}(\boldsymbol{y} - H\boldsymbol{x})$. DMSP estimates the noise level $\Sigma$ by maximizing (S6) for $\Sigma$. If $\Sigma$ is isotropic, we see that the maximizer is

$$\Sigma^* = \frac{1}{d}\big[\|\boldsymbol{y} - H\boldsymbol{x}\|^2 + \sigma^2 \text{trace}(H H^\top)\big] I, \tag{S7}$$

where $d$ is the dimension of $\boldsymbol{y}$. This maximizer coincides with (21) in [*S2*]. It is shown to overestimate the true noise level by the experimental results in the main paper, as the data term is based on a

Gaussian smoothed log-likelihood, which introduces a correction term $\sigma^2\text{trace}(HH^\top)$ to compensate for the overfitting caused by using a single sample to estimate $\Sigma$. This overestimation phenomenon is especially significant if the smoothing kernel $\sigma$ is close to the true noise level.

If multiple samples $\boldsymbol{y}_1, ..., \boldsymbol{y}_k$ share the same noise level in DMSP, the data term is simply the sum of individual data terms, as samples are mutually independent conditioned on $\Sigma$:

$$
\begin{aligned}
\text{data}(\{\boldsymbol{x}_i\}_{i=1}^k) = &-\frac{1}{2}\sum_{i=1}^k (\boldsymbol{y}_i - H\boldsymbol{x}_i)^\top \Sigma^{-1}(\boldsymbol{y}_i - H\boldsymbol{x}_i) \\
&-\frac{k}{2}\sigma^2\text{trace}(\Sigma^{-1}HH^\top) - \frac{k}{2}\log|\Sigma| + const.,
\end{aligned}
\tag{S8}
$$

the maximizer can be similarly solved. If $\Sigma$ is constrained to be diagonal instead of isotropic, the diagonal elements of $\Sigma$ can be solved using (S8), by noticing that the trace of $\Sigma^{-1}HH^\top$ is simply the weighted sum of the diagonal elements of $HH^\top$, weighted by the diagonal elements of $\Sigma^{-1}$.

## A.2 DAEP

DAEP requires a known noise level $\Sigma = \sigma_d^2 I$. It is used to compute the data term gradient during gradient-based optimization (page 4 of [S1])

$$
\nabla_{\boldsymbol{x}} L(\boldsymbol{x} \mid \boldsymbol{y}) = H^\top(H\boldsymbol{x} - \boldsymbol{y})/\sigma_d^2,
\tag{S9}
$$

To use a full general covariance $\boldsymbol{n} \sim \mathcal{N}(0, \Sigma)$, we see that the negative log-likelihood $L(\boldsymbol{x} \mid \boldsymbol{y})$ now satisfies

$$
L(\boldsymbol{x} \mid \boldsymbol{y}) = \frac{1}{2}(H\boldsymbol{x} - \boldsymbol{y})^\top \Sigma^{-1}(H\boldsymbol{x} - \boldsymbol{y}) + const.
\tag{S10}
$$

therefore we can simply replace (S9) with

$$
\nabla_{\boldsymbol{x}} L(\boldsymbol{x} \mid \boldsymbol{y}) = H^\top \Sigma^{-1}(H\boldsymbol{x} - \boldsymbol{y}).
\tag{S11}
$$

## A.3 ADMM

ADMM also requires a known noise level $\Sigma$. It is used for specifying the x-subproblem

$$
\boldsymbol{x}^{(k+1)} = \text{argmax}_{\boldsymbol{x}} \ \log \Pr(\boldsymbol{y} \mid \boldsymbol{x}) - \frac{\lambda}{2}\|\boldsymbol{x} - \boldsymbol{v}^{(k)} + \boldsymbol{u}^{(k)}\|^2
\tag{S12}
$$

$$
= \text{argmin}_{\boldsymbol{x}} \ (H\boldsymbol{x} - \boldsymbol{y})^\top \Sigma^{-1}(H\boldsymbol{x} - \boldsymbol{y}) + \lambda(\boldsymbol{x} - \tilde{\boldsymbol{x}}^{(k)})^\top(\boldsymbol{x} - \tilde{\boldsymbol{x}}^{(k)}),
\tag{S13}
$$

where we define $\tilde{\boldsymbol{x}}^{(k)} = \boldsymbol{v}^{(k)} - \boldsymbol{u}^{(k)}$. The solution to (S13) can easily be derived as

$$
\boldsymbol{x}^{(k+1)} = (H^\top \Sigma^{-1}H + \lambda I)^{-1}(H^\top \Sigma^{-1}\boldsymbol{y} + \lambda \tilde{\boldsymbol{x}}^{(k)}).
\tag{S14}
$$

# B   More on the M-step

In the Monte Carlo EM [S5] algorithm, the M-step computes the following objective:

$$
\Sigma^* = \text{argmax}_{\Sigma} \ \sum_{i=1}^n \log \Pr(\boldsymbol{y} \mid \boldsymbol{x}^{(i)}, \Sigma),
\tag{S15}
$$

where the log-likelihood is

$$
\log \Pr(\boldsymbol{y} \mid \boldsymbol{x}, \Sigma) = -\tfrac{1}{2}(\boldsymbol{y} - H\boldsymbol{x})^\top \Sigma^{-1}(\boldsymbol{y} - H\boldsymbol{x}) - \tfrac{1}{2}\log|\Sigma| + const.
\tag{S16}
$$

The general solution is

$$
\Sigma^* = \frac{1}{n}\sum_{i=1}^n (\boldsymbol{y} - H\boldsymbol{x}^{(i)})(\boldsymbol{y} - H\boldsymbol{x}^{(i)})^\top.
\tag{S17}
$$

Here we discuss some special cases:

1. The noise level $\Sigma$ is constrained to be isotropic.
2. The noise level $\Sigma$ is constrained to be diagonal.
3. Multiple observations $\boldsymbol{y}$ share the same $\Sigma$.

If the noise level $\Sigma$ is isotropic, we parametrize $\Sigma$ as $\sigma I$, where $I$ is the $d \times d$ identity matrix, $d$ is the dimension of $\boldsymbol{y}$. It is straightforward to derive from (S15) and (S16) that

$$\sigma^* = \frac{1}{nd} \sum_{i=1}^{n} (\boldsymbol{y} - H\boldsymbol{x}^{(i)})^\top (\boldsymbol{y} - H\boldsymbol{x}^{(i)}). \tag{S18}$$

If the noise level is diagonal, we parametrize $\Sigma$ as $\mathrm{diag}(\sigma_1, ..., \sigma_d)$. Each dimension can be treated independently, and we see from (S15) and (S16) that

$$\sigma_k^* = \frac{1}{n} \sum_{i=1}^{n} (\boldsymbol{y}_{[k]} - H_k \boldsymbol{x}^{(i)})^2, k = 1, ..., d, \tag{S19}$$

where $\boldsymbol{y}_{[k]}$ is the $k$-the coordinate of $\boldsymbol{y}$, $H_k$ is the $k$-th row of matrix $H$.

If we have multiple observations $\boldsymbol{y}_1, ..., \boldsymbol{y}_k$ that share the same $\Sigma$, conditioned on $\Sigma$ they are mutually independent. The expected complete log-likelihood becomes

$$\mathcal{Q}(\Sigma, \Sigma^{(\tau)}) = \sum_{j=1}^{k} \mathbb{E}_{\boldsymbol{x}_j \sim \mathrm{Pr}(\boldsymbol{x}_j | \boldsymbol{y}_j, \Sigma^{(\tau)})} \log \mathrm{Pr}(\boldsymbol{y}_j, \boldsymbol{x}_j \mid \Sigma)$$

$$= \sum_{j=1}^{k} \mathbb{E}_{\boldsymbol{x}_j \sim \mathrm{Pr}(\boldsymbol{x}_j | \boldsymbol{y}_j, \Sigma^{(\tau)})} \log \mathrm{Pr}(\boldsymbol{y}_j \mid \boldsymbol{x}_j, \Sigma) + \log \mathrm{Pr}(\boldsymbol{x}_j). \tag{S20}$$

During the Monte Carlo E-step, we sample $n$ samples for each $\boldsymbol{x}_j$, resulting in a total of $nk$ samples $\{\boldsymbol{x}_j^{(i)}\}_{i=1}^{n}, j = 1, ..., k$. During the M-step, the objective (S15) is replaced by

$$\Sigma^* = \mathrm{argmax}_\Sigma \sum_{j=1}^{k} \sum_{i=1}^{n} \log \mathrm{Pr}(\boldsymbol{y}_j \mid \boldsymbol{x}_j^{(i)}, \Sigma). \tag{S21}$$

## C    More on Metropolis-adjusted Langevin algorithm

The Metropolis-adjusted Langevin algorithm (MALA) [*S3*] is a Metropolis-Hastings sampler with a specially chosen proposal distribution. In general, in order to sample from a target distribution $\pi(\cdot)$, an Metropolis-Hastings sampler draws a value $x^*$ from a proposal distribution $q(\cdot \mid x)$, based on the current value $x$. The proposed value $x^*$ is accepted with probability

$$\min \left(1, \frac{\pi(x^*)q(x \mid x^*)}{\pi(x)q(x^* \mid x)}\right). \tag{S22}$$

The MALA uses the following proposal distribution

$$x^* \sim \mathcal{N}(x + \frac{\sigma^2}{2} \nabla \log \pi(x), \sigma^2). \tag{S23}$$

The intuition of MALA comes from the Langevin diffusion, which is based on the following stochastic differential equation (SDE):

$$\mathrm{d}X_t = \nabla f(X_t) \, \mathrm{d}t + \sqrt{2} \, \mathrm{d}B_t, \tag{S24}$$

where $B_t$ is the standard Brownian motion, the function $f$ is the energy of the target distribution $\pi(x) = Z^{-1} \exp(-f(x))$, where $Z$ is a normalization constant. Under mild conditions, the solution to the SDE is an ergodic Markov process whose unique stationary distribution is $\pi$. Therefore, we can use discretized simulation of the SDE to sample from the stationary distribution $\pi$, by the following recursion:

$$X_{n+1} = X_n + \delta \nabla f(X_n) + \sqrt{2\delta} \epsilon, \tag{S25}$$

where $\delta$ is a constant and $\epsilon$ is a standard normal random variable. Due to the error introduced during discretization, a Metropolis-Hastings accept/reject step is added for correction. This gives the MALA as in (S23).

Figure S1: Projection of the 2D manifold (S26) on the $k$-th and $k + 1$-th coordinates, where $k = 1, ..., 49$. It can be seen that this manifold is highly nonlinear.

## D Visualization of simulated data

In the signal denoising experiment we considered a hypothetical 2D manifold in a 50-dimensional space, which can be parametrized by two uniform random variables as follows,

$$x_k = 0.01(\alpha + \beta)^2 \sin[\alpha \sin(ke) + \beta \sin(ke + 1) + 0.5(\alpha + \beta)], k = 1, ..., 50, \quad \text{(S26)}$$

where $\alpha, \beta \sim \text{Uniform}(2, 5)$, $e = \exp(1)$ is the Euler constant, and $x_k$ is the $k$-th coordinate of the 50-dimensional signal. Here we provide visualization of this manifold and show that it is highly nonlinear. Fig. S1 shows the projection of this manifold on the $k$-th and $k + 1$-th coordinates, where $k = 1, ..., 49$.

## E More experimental results

Due to the space limit of the main paper, here we present the result of an additional experiment, where the generalized baseline methods from Section A are compared with our proposed methods.

**Time series deconvolution.** In this experiment we consider a non-invertible transform $H$, as well as multiple-$\boldsymbol{y}$ analysis (i.e. $k > 1$). Suppose a time series $\boldsymbol{x} = \{x_1, ..., x_{10}\}$ is convoluted with kernel

Table S1: Average RMSE for time series deconvolution on the test set. Standard deviations are in parentheses, the best performance is in **bold**. (*All values are in* $10^{-2}$).

| Method: | DAEP+NE | ADMM+NE | DMSP | AGEM | AGEM-ADMM |
|---|---|---|---|---|---|
| mean | 1.10 | 0.84 | 0.91 | 0.72 | **0.67** |
| std. | (0.34) | (0.66) | (0.56) | (0.39) | (0.52) |

Table S2: Estimated noise levels for time series deconvolution. (*All values are in* $10^{-2}$)

| Method | | $\Sigma_{11}$ | $\Sigma_{22}$ | $\Sigma_{33}$ | $\Sigma_{44}$ | $\Sigma_{55}$ | $\Sigma_{66}$ | $\Sigma_{77}$ | $\Sigma_{88}$ |
|---|---|---|---|---|---|---|---|---|---|
| Method | true: | 1.00 | 2.00 | 3.00 | 4.00 | 1.00 | 2.00 | 3.00 | 4.00 |
| DMSP | mean | 1.62 | 2.31 | 2.77 | 3.78 | 1.83 | 2.14 | 2.98 | 3.90 |
| | std. | (0.07) | (0.47) | (0.67) | (0.53) | (0.13) | (0.24) | (0.28) | (0.45) |
| AGEM | mean | 1.09 | 2.14 | 2.70 | 3.73 | 1.27 | 1.83 | 2.88 | 3.91 |
| | std. | (0.01) | (0.03) | (0.02) | (0.01) | (0.01) | (0.01) | (0.03) | (0.02) |

(-0.33, 1.0, -0.33) using VALID padding, then corrupted with temporal variant and independent noise (i.e. $\Sigma$ is diagonal). Here we set the dimension of $\boldsymbol{x}$ and $\boldsymbol{y}$ to small values for better display the estimated noise level. Further suppose that multiple observations arrive at the same time, so that we can use them to jointly estimate the noise level. We generate 5000 time series from a latent 1D manifold,

$$x_k = 0.01\alpha^2 \sin[\alpha \sin(ke) + 0.5\alpha], k = 1, ..., 10. \tag{S27}$$

where $\alpha \sim \text{Uniform}(2, 5)$, $e = \exp(1)$. This is simply a slice ($\beta = 0$) of the 2D manifold (S26). Among 5000 samples, 250 samples are selected as the validation set and another 250 samples as the test set. The rest are used for DAE training. DAE architecture and training follow the signal denoising experiment in the main paper, except that now each hidden layer contains 500 neurons instead of 2000, since the data dimension is reduced.

For testing, the linear transformation $H$ is the $8 \times 10$ Toeplitz matrix of the convolution kernel, which is non-invertible. We consider the diagonal noise $\Sigma = 0.01\text{diag}(1, 2, 3, 4, 1, 2, 3, 4)$. The validation set is grouped into 10 cases, each case contains 25 observations for joint estimation. The same is done to the test set. We set $n_{\text{EM}} = 30$, $n_{\text{MH}} = 1000$. The hyper-parameter $\sigma_{\text{prop}}$ is set to 0.008 using the same selection method as signal denoising. RMSE and estimated noise levels, averaged over 10 test cases of the test set, are reported in Table S1 and S2. We see that both our methods outperform all baselines statistically significantly ($p < 0.01$) in terms of RMSE, and the noise estimator of AGEM has much lower variance than that of DMSP. This is due to MH's ability to better explore the posterior distribution. In contrast, DMSP estimates the noise level only based on the current iteration of $\boldsymbol{x}$, which is especially problematic in this experiment setting (only 25 samples per dimension).