[Reviews · NeurIPS 2019]

Reviewer 1



The paper deals with inverse problems addressed within the Bayesian framework. More precisely, the prior for the unknown object is defined through a given set of candidate objects (data-driven prior). The paper proposes a solution for the question of hyperparameter tuning and it is original regarding this point. It clearly propose an advance of the state of the art for a class of important problems. The paper is technically sound and clear written.

Reviewer 2



Quality: The overall approach proposed by the paper seems to be technically sound. One weakness of the proposed AGEM is that the result is sensitive to the sigma in the proposal distribution in MH-E step. I wonder if the authors tried other MCMC sampling algorithms, such as HMC, or adaptive MH proposals (such as Haario et al 2001). Overall, the paper did a great job of clearly stating the experimental details, make the results reproducible. Clarity: The paper well-written and easy to follow. The details of the datasets and experiments are well documented. Originality: The proposed AGEM algorithm for simultaneously solving linear inverse problem and estimating noise level is novel to my knowledge. References: Haario, H., Saksman, E., & Tamminen, J. (2001). An adaptive Metropolis algorithm. Bernoulli, 7(2), 223-242.

Reviewer 3



Response to authors: I thank the authors for their answers. I appreciate the additional results provided by the authors, which shows the stability of their approach. I still think that the paper needs more work for publication (see below). However, after discussions with other reviewers, I will upgrade my original suggestion. * Currently the paper does not show very-well the contributions, and does not put them in perspective w.r.t. prior work (MMSE estimators and noise estimation methods). * As the authors mention, the paper proposes a new sampling technique for the posteriori: We should be able to see some examples of these samples see if they do have a "visual" quality. * The sampling quality is not very well supported by the proposed method. The ADMM approach (which is build on top of MAP) achieves better PSNR values in average compared to the MMSE solution. This shows that the method is not performing well (at least compared to MAP solutions). I do not expect that the method performs well here, but I think it is necessary that we see the evidence and hear what the authors think and investigate in this regard. ----------------------------- This paper presents a method for sampling a posterior distribution using DAEs for solving linear inverse problems. The sampling technique is based on two approximations: kernel density estimation as in DMSP approach, and linear approximation of the density (in Log) for sampling. The paper is well written and the proposed technique is novel. I have, however, one major concern about the theoretical explanations and experiments. The proposed method AGEM, is not a MAP estimator as far as I understood: averaging samples from the posteriori leads to MMSE solution and not the most probable (MAP). This, however is not described at the begging (Eq. 2) and presented as a MAP solution. I think that having a Minimum Mean Squared Error (MMSE) estimator is, by itself, very interesting and a good contribution. But the presentation of the method shows that the authors did not consider this fact. I ask the authors to please address this issue and clarify if this is not the case. Putting this method in context, the experiments and results are missing more important comparisons to other MMSE estimators. Summary of the points: 0) The AGEM method is an MMSE estimator, not MAP. 1) Equation 15 seems to be missing the q() terms from Eq. 14, please clarify. 2) Linear approximation in 16 needs to be discussed and clarified. What are the consequences? Alain and Bengio use similar approximations by taking a summation over interpolated samples. How do the two approaches compare? 3) I think the paper is missing a convergence visualization (change in results/PSNR) of AGEM with different number of samples.

[Author Response · NeurIPS 2019]

We thank the reviewers for their valuable comments. Raised issues are addressed below.

=== *Reviewer #1* === **Q1: Notations improvement.** We thank R1 for the advice and will revise the paper accordingly.

=== *Reviewer #2* === **Q2: I wonder if the authors tried other MCMC sampling algorithms.** We tested two
MCMC sampling algorithms: the original MH with Gaussian proposals (Table 2, last row and column), and MALA.
Other variations may also be viable if related quantities can be evaluated with DAEs. We thank R2 for the suggestion.

**Q3: Compare the performance of AGEM with GAN priors.** As our main goal is to improve existing DAE prior
based methods, we compared with previous work using priors defined by the same DAE. However, as we mentioned in
the future work, it is also very interesting to study other deep priors. We thank R2 for pointing out this direction.

=== *Reviewer #3* === **Q4: Averaging samples from the posterior leads to MMSE solution and not MAP. This,**
**however is not described at the beginning (Eq. 2) and presented as an MAP solution. Is the method an MAP or**
**an MMSE estimator?** It seems that the reviewer might have misunderstood part of our paper. The MAP formulation
(Eq. 2) only serves to explain MAP-based previous methods (DAEP and ADMM), it is never used in our method. The
goal of this paper IS NOT to improve existing MAP/MMSE solution, but to improve existing methods that use DAE
priors, as we are interested in a better solution that is plug-and-play and only requires unsupervised training. (see Q5)

Our result, the ability to sample from the posterior, is more general than a point estimator, because the latter can
be constructed from the samples. As line 125 notes, "a simple choice is to use their average as the recovered $\hat{x}$."
The reviewer is correct in pointing out that this leads to an MMSE estimator. The primary reason for doing so is
computational: in Table 2, row "misc", we compared the posterior mean and median. Their performances are close,
but the mean is easier to compute. Another reason is that many applications care about MSE (e.g. PSNR for images),
hence MMSE estimator is arguably more suitable. In principle, the Bayes estimator of common loss functions can be
constructed according to Bayesian decision theory (e.g. posterior mean for MSE, posterior median for L1 loss), our
method is not restricted to any particular loss function.

**Q5: Give precise definition of the objective.** As our abstract indicates (also summarized by R2), our objective is to
improve existing work that use DAE prior for solving linear inverse problems (Eq. 1). The solution can be derived
using different estimators: DAEP and ADMM use MAP, DMSP proposes a Bayes estimator of a special utility function
[There was a typo near line 69: "MAP solution" should be just "solution" as DMSP is not an MAP estimator, as is clear
from their paper's abstract], while we use MMSE. Again, there is no restriction on what estimator is used, as long as
they perform well in actual tasks (e.g. obtain good PSNR on image restoration). This is the same criteria as the DMSP
paper. The reason to focus on DAE priors is explained in line 54-56 (also in line 14 and 35 of this rebuttal).

**Q6: Compare to relevant methods, missing comparisons to other MMSE estimators.** As this paper focuses on
DAE prior, we primarily compare with existing methods DMSP/DAEP/ADMM (using the same DAE). Among these,
we are the first to be able to provide MMSE estimator. MMSE is not the main theme of comparison according to our
objective (Q5), plus the posterior median also works well (Table 2). As for non-DAE prior, they are compared in Table
3, 4, also in the DAEP/DMSP paper. It is clear that DAE based methods have competitive performance, and their main
advantage lies in plug-and-play (does not rely on task), have minimal handcrafting and fully unsupervised training.

**Q7: Equation 15 seems to be missing the q() terms from Eq. 14, please clarify.** As noted in line 108 and 104, in
Section 3.1 we first illustrate our method using a simple, symmetric Gaussian proposal $q$, hence in Eq. 15 two q factors
cancel. Only in Section 3.2 is the MALA introduced, the proposal then becomes asymmetric.

**Q8: Linear approximation in 16 needs to be discussed and clarified. How does the approach compare to**
**Alain and Bengio?** As mentioned in Alain and Bengio, multiple-step discretization can be used if more accurate
approximation is needed. We did try the multiple-step method, however, the difference in results turns out to be
marginal. The main time cost of our approach is due to the evaluation of reconstructed error using the DAE (Eq. 19).
Therefore, using an $m$-step approximation would roughly $m\times$ the total time cost. For practical applications, we found
the one-step approximation to be cost-efficient.

**Q9: Missing a convergence visualization.** We thank R3 for the suggestion. Please see figure below for an example.

Figure 1: AGEM for image deblurring (Table 3). Left: average estimated noise level; right: mean PSNR. The legend
shows the true noise level $\sigma_n$. Stable convergence is quickly reached. Each EM epoch draws 300 MCMC samples.

[Meta-Review · NeurIPS 2019]

Congratulations, your paper has been accepted for publication at NeurIPS2019. The reviewers found it to be an important and novel piece of work. When preparing the camera ready version, please bear in mind the reviewers comments. In particular, the following were raised during the discussion as outstanding points that should be considered when revising the paper: - Clarify whether AGEM is a MAP or a MMSE estimator. All three reviewers insisted upon this point during discussion. - Include the additional results from your response which show the stability of your approach. - Define more carefully what the main contribution of the paper is, and place it in perspective wrt prior work (ie MMSE estimators and noise estimation methods).